# Coherent Hierarchical Multi-Label Classification Networks

**Eleonora Giunchiglia**
Department of Computer Science
University of Oxford, UK
eleonora.giunchiglia@cs.ox.ac.uk

**Thomas Lukasiewicz**
Department of Computer Science
University of Oxford, UK
thomas.lukasiewicz@cs.ox.ac.uk

## Abstract

Hierarchical multi-label classification (HMC) is a challenging classification task extending standard multi-label classification problems by imposing a hierarchy constraint on the classes. In this paper, we propose C-HMCNN($h$), a novel approach for HMC problems, which, given a network $h$ for the underlying multi-label classification problem, exploits the hierarchy information in order to produce predictions coherent with the constraint and improve performance. We conduct an extensive experimental analysis showing the superior performance of C-HMCNN($h$) when compared to state-of-the-art models.

## 1 Introduction

Multi-label classification is a standard machine learning problem in which an object can be associated with multiple labels. A hierarchical multi-label classification (HMC) problem is defined as a multi-label classification problem in which classes are hierarchically organized as a tree or as a directed acyclic graph (DAG), and in which every prediction must be coherent, i.e., respect the hierarchy constraint. The *hierarchy constraint* states that a datapoint belonging to a given class must also belong to all its ancestors in the hierarchy. HMC problems naturally arise in many domains, such as image classification [12–14], text categorization [17, 20, 27], and functional genomics [1, 9, 32]. They are very challenging for two main reasons: (i) they are normally characterized by a great class imbalance, because the number of datapoints per class is usually much smaller at deeper levels of the hierarchy, and (ii) the predictions must be coherent. Consider, e.g., the task proposed in [13], where a radiological image has to be annotated with an IRMA code, which specifies, among others, the biological system examined. In this setting, we expect to have many more "abdomen" images than "lung" images, making the label "lung" harder to predict. Furthermore, the prediction "respiratory system, stomach" should not be possible given the hierarchy constraint stating that "stomach" belongs to "gastrointestinal system". While most of the proposed methods directly output predictions that are coherent with the hierarchy constraint (see, e.g., [3, 22]), there are models that allow incoherent predictions and, at inference time, require an additional post-processing step to ensure its satisfaction (see, e.g., [6, 24, 31]). Most of the state-of-the-art models based on neural networks belong to the second category (see, e.g., [6, 7, 33]).

In this paper, we propose C-HMCNN($h$), a novel approach for HMC problems, which, given a network $h$ for the underlying multi-label classification problem, exploits the hierarchy information to produce predictions coherent with the hierarchy constraint and improve performance. C-HMCNN($h$) is based on two basic elements: (i) a constraint layer built on top of $h$, ensuring that the predictions are coherent by construction, and (ii) a loss function teaching C-HMCNN($h$) when to exploit the prediction on the lower classes in the hierarchy to make predictions on the upper ones. C-HMCNN($h$) has the following four features: (i) its predictions are coherent without any post-processing, (ii) differently from other state-of-the-art models (see, e.g., [33]), its number of parameters is independent from the number of hierarchical levels, (iii) it can be easily implemented on GPUs using standard

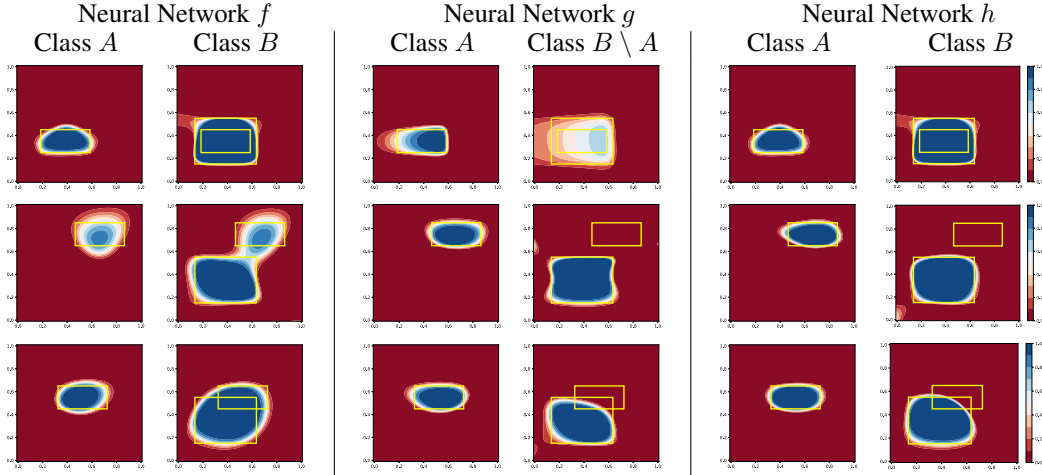

| Neural Network $f$ | | Neural Network $g$ | | Neural Network $h$ | |
| Class $A$ | Class $B$ | Class $A$ | Class $B \setminus A$ | Class $A$ | Class $B$ |

Figure 1: In all figures, the smaller yellow rectangle corresponds to $R_1$, while the bigger yellow one corresponds to $R_2$. The first row of figures corresponds to $R_1 \cap R_2 = R_1$, the second corresponds to $R_1 \cap R_2 = \emptyset$, and the third corresponds to $R_1 \cap R_2 \notin \{R_1, \emptyset\}$. First 4 columns: decision boundaries of $f$ (resp., $g$) for classes $A$ and $B$ (resp., $A$ and $B \setminus A$). Last 2 columns: decision boundaries of $h$ for classes $A$ and $B$. In each figure, the darker the blue (resp., red), the more confident a model is that the datapoints in the region belong (do not belong) to the class (see the scale at the end of each row).

libraries, and (iv) it outperforms the state-of-the-art models Clus-Ens [28], HMC-LMLP [7], HMCN-F, and HMCN-R [33] on 20 commonly used real-world HMC benchmarks.

The rest of this paper is organized as follows. In Section 2, we introduce the notation and terminology used. Then, in Section 3, we present the core ideas behind C-HMCNN($h$) on a simple HMC problem with just two classes, followed by the presentation of the general solution in Section 4. Experimental results are presented in Section 5, while the related work is discussed in Section 6. The last section gives some concluding remarks.

## 2 Notation and terminology

Consider an arbitrary HMC problem with a given set of classes, which are hierarchically organized as a DAG. If there is a path of length $\geq 0$ from a class $A$ to a class $B$ in the DAG, then we say that $B$ is a *subclass* of $A$ (every class is thus a subclass of itself). Consider an arbitrary datapoint $x \in \mathbb{R}^D$, $D \geq 1$. For each class $A$ and model $m$, we assume to have a mapping $m_A \colon \mathbb{R}^D \to [0, 1]$ such that $x \in \mathbb{R}^D$ is predicted to belong to $A$ whenever $m_A(x)$ is bigger than or equal to a user-defined threshold. To guarantee that the hierarchy constraint is always satisfied independently from the threshold, the model $m$ should guarantee that $m_A(x) \leq m_B(x)$, for all $x \in \mathbb{R}^D$, whenever $A$ is a subclass of $B$: if $m_A(x) > m_B(x)$, for some $x \in \mathbb{R}^D$, then we have a *hierarchy violation* (see, e.g., [33]). For ease of readability, in the rest of the paper, we always leave implicit the dependence on the considered datapoint $x$, and write, e.g., $m_A$ for $m_A(x)$.

## 3 Basic case

Our goal is to leverage standard neural network approaches for multi-label classification problems and then exploit the hierarchy constraint in order to produce coherent predictions and improve performance. Given our goal, we first present two basic approaches, exemplifying their respective strengths and weaknesses. These are useful to then introduce our solution, which is shown to present their advantages without exhibiting their weaknesses. In this section, we assume to have just two classes $A \subseteq \mathbb{R}^D$ and $B \subseteq \mathbb{R}^D$ and the constraint stating that $A$ is a subclass of $B$.

### 3.1 Basic approaches

In the first approach, we treat the problem as a standard multi-label classification problem and simply set up a neural network $f$ with one output per class to be learnt: to ensure that no hierarchy violation

happens, we need an additional post-processing step. In this simple case, the post-processing could set the output for $A$ to be $\min(f_A, f_B)$ or the output for $B$ to be $\max(f_B, f_A)$. In this way, all predictions are always coherent with the hierarchy constraint. Another approach for this case is to build a network $g$ with two outputs, one for $A$ and one for $B \setminus A$. To meaningfully ensure that no hierarchy violation happens, we need an additional post-processing step in which the predictions for the class $B$ are given by $\max(g_{B \setminus A}, g_A)$. Considering the two above approaches, depending on the specific distribution of the points in $A$ and in $B$, one solution may be significantly better than the other, and a priori we may not know which one it is.

To visualize the problem, assume that $D = 2$, and consider two rectangles $R_1$ and $R_2$ with $R_1$ smaller than $R_2$, like the two yellow rectangles in the subfigures of Figure 1. Assume $A = R_1$ and $B = R_1 \cup R_2$. Let $f^+$ be the model obtained by adding a post-processing step to $f$ setting $f_A^+ = \min(f_A, f_B)$ and $f_B^+ = f_B$, as in [6, 7, 15] (analogous considerations hold, if we set $f_A^+ = f_A$ and $f_B^+ = \max(f_B, f_A)$ instead). Intuitively, we expect $f^+$ to perform well even with a very limited number of neurons when $R_1 \cap R_2 = R_1$, as in the first row of Figure 1. However, if $R_1 \cap R_2 = \emptyset$, as in the second row of Figure 1, we expect $f^+$ to need more neurons to obtain the same performance. Consider the alternative network $g$, and let $g^+$ be the system obtained by setting $g_A^+ = g_A$ and $g_B^+ = \max(g_{B \setminus A}, g_A)$. Then, we expect $g^+$ to perform well when $R_1 \cap R_2 = \emptyset$. However, if $R_1 \cap R_2 = R_1$, we expect $g^+$ to need more neurons to obtain the same performance. We do not consider the model with one output for $B \setminus A$ and one for $B$, since it performs poorly in both cases.

To test our hypothesis, we implemented $f$ and $g$ as feedforward neural networks with one hidden layer with 4 neurons and *tanh* nonlinearity. We used the *sigmoid* non-linearity for the output layer (from here on, we always assume that the last layer of each neural network presents *sigmoid* non-linearity). $f$ and $g$ were trained with binary cross-entropy loss using Adam optimization [16] for 20k epochs with learning rate $10^{-2}$ ($\beta_1 = 0.9, \beta_2 = 0.999$). The datasets consisted of 5000 (50/50 train test split) datapoints sampled from a uniform distribution over $[0, 1]^2$. The first four columns of Figure 1 show the decision boundaries of $f$ and $g$. Those of $f^+$ and $g^+$, reported in Appendix A, can be derived from the plotted ones, while the converse does not hold. These figures highlight that $f$ (resp., $g$) approximates the two rectangles better than $g$ (resp., $f$) when $R_1 \cap R_2 = R_1$ (resp., $R_1 \cap R_2 = \emptyset$). In general, when $R_1 \cap R_2 \notin \{R_1, \emptyset\}$, we expect that the behavior of $f$ and $g$ depends on the relative position of $R_1$ and $R_2$.

## 3.2 Our solution

Ideally, we would like to build a neural network that is able to have roughly the same performance of $f^+$ when $R_1 \cap R_2 = R_1$, of $g^+$ when $R_1 \cap R_2 = \emptyset$, and better than both in any other case. We can achieve this behavior in two steps. In the first step, we build a new neural network consisting of two modules: (i) a bottom module $h$ with two outputs in $[0, 1]$ for $A$ and $B$, and (ii) an upper module, called *max constraint module (MCM)*, consisting of a single layer that takes as input the output of the bottom module and imposes the hierarchy constraint. We call the obtained neural network *coherent hierarchical multi-label classification neural network* (C-HMCNN($h$)).

Consider a datapoint $x$. Let $h_A$ and $h_B$ be the outputs of $h$ for the classes $A$ and $B$, respectively, and let $y_A$ and $y_B$ be the ground truth for the classes $A$ and $B$, respectively.

The outputs of MCM (which are also the output of C-HMCNN($h$)) are:

$$
\begin{aligned}
\text{MCM}_A &= h_A, \\
\text{MCM}_B &= \max(h_B, h_A).
\end{aligned}
\tag{1}
$$

Notice that the output of C-HMCNN($h$) ensures that no hierarchy violation happens, i.e., that for any threshold, it cannot be the case that MCM predicts that a datapoint belongs to $A$ but not to $B$. In the second step, to exploit the hierarchy constraint during training, C-HMCNN($h$) is trained with a novel loss function, called *max constraint loss (MCLoss)*, defined as $\text{MCLoss} = \text{MCLoss}_A + \text{MCLoss}_B$, where:

$$
\begin{aligned}
\text{MCLoss}_A &= -y_A \ln(\text{MCM}_A) - (1 - y_A)\ln(1 - \text{MCM}_A), \\
\text{MCLoss}_B &= -y_B \ln(\max(h_B, h_A y_A)) - (1 - y_B)\ln(1 - \text{MCM}_B)).
\end{aligned}
\tag{2}
$$

MCLoss differs from the standard binary cross-entropy loss

$$
\mathcal{L} = -y_A \ln(\text{MCM}_A) - (1 - y_A)\ln(1 - \text{MCM}_A) - y_B \ln(\text{MCM}_B) - (1 - y_B)\ln(1 - \text{MCM}_B),
$$

iff $x \notin A$ ($y_A = 0$), $x \in B$ ($y_B = 1$), and $h_A > h_B$.

The following example highlights the different behavior of MCLoss compared to $\mathcal{L}$.

**Example 3.1.** Assume $h_A = 0.3$, $h_B = 0.1$, $y_A = 0$, and $y_B = 1$. Then, we obtain:

$$\mathcal{L} = -\ln(1 - \text{MCM}_A) - \ln(\text{MCM}_B) = -\ln(1 - h_A) - \ln(h_A).$$

Given the above, we get:

$$\frac{\partial \mathcal{L}}{\partial h_A} = -\frac{1}{h_A - 1} - \frac{1}{h_A} \sim -1.9 \qquad \frac{\partial \mathcal{L}}{\partial h_B} = 0.$$

Hence, if C-HMCNN($h$) is trained with $\mathcal{L}$, then it wrongly learns that it needs to increase $h_A$ and keep $h_B$. On the other hand, for C-HMCNN($h$) (with MCLoss), we obtain:

$$\frac{\partial \text{MCLoss}}{\partial h_A} = -\frac{1}{h_A - 1} \sim 1.4 \qquad \frac{\partial \text{MCLoss}}{\partial h_B} = -\frac{1}{h_B} = -10.$$

In this way, C-HMCNN($h$) rightly learns that it needs to decrease $h_A$ and increase $h_B$.

Consider the example in Figure 1. To check that our model behaves as expected, we implemented $h$ as $f$, and trained C-HMCNN($h$) with MCLoss on the same datasets and in the same way as $f$ and $g$. The last two columns of Figure 1 show the decision boundaries of $h$ (those of C-HMCNN($h$) can be derived from the plotted ones and are in Appendix A). $h$'s decision boundaries mirror those of $f$ (resp., $g$) when $R_1 \cap R_2 = R_1$ (resp., $R_1 \cap R_2 = \emptyset$). Intuitively, C-HMCNN($h$) is able to decide whether to learn $B$: (i) as a whole (top figure), (ii) as the union of $B \setminus A$ and $A$ (middle figure), and (iii) as the union of a subset of $B$ and a subset of $A$ (bottom figure). C-HMCNN($h$) has learnt when to exploit the prediction on the lower class $A$ to make predictions on the upper class $B$.

## 4 General case

Consider a generic HMC problem with a set $\mathcal{S}$ of $n$ hierarchically structured classes, a datapoint $x \in \mathbb{R}^D$, and a generic neural network $h$ with one output for each class in $\mathcal{S}$. Given a class $A \in \mathcal{S}$, $\mathcal{D}_A$ is the set of subclasses of $A$ in $\mathcal{S}$.[1] $y_A$ is the ground truth label for class $A$ and $h_A \in [0, 1]$ is the prediction made by $h$ for $A$. The output $\text{MCM}_A$ of C-HMCNN($h$) for a class $A$ is:

$$\text{MCM}_A = \max_{B \in \mathcal{D}_A} (h_B). \tag{3}$$

For each class $A \in \mathcal{S}$, the number of operations performed by $\text{MCM}_A$ is independent from the depth of the hierarchy, making C-HMCNN($h$) a scalable model. Thanks to MCM, C-HMCNN($h$) is guaranteed to always output predictions satisfying the hierarchical constraint, as stated by the following theorem, which follows immediately from Eq. (3).

**Theorem 4.1.** *Let $\mathcal{S} = \{A_1, \ldots, A_n\}$ be a set of hierarchically structured classes. Let $h$ be a neural network with outputs $h_{A_1}, \ldots, h_{A_n} \in [0, 1]$. Let $MCM_{A_1}, \ldots, MCM_{A_n}$ be defined as in Eq. (3). Then, C-HMCNN($h$) does not admit hierarchy violations.*

For each class $A \in \mathcal{S}$, $\text{MCLoss}_A$ is defined as:

$$\text{MCLoss}_A = -y_A \ln(\max_{B \in \mathcal{D}_A} (y_B h_B)) - (1 - y_A) \ln(1 - \text{MCM}_A).$$

The final MCLoss is given by:

$$\text{MCLoss} = \sum_{A \in \mathcal{S}} \text{MCLoss}_A. \tag{4}$$

The importance of using MCLoss instead of the standard binary cross-entropy loss $\mathcal{L}$ becomes even more apparent in the general case. Indeed, as highlighted by the following example, the more ancestors a class has, the more likely it is that C-HMCNN($h$) trained with $\mathcal{L}$ will remain stuck in bad local optima.

**Example 4.2.** Consider a generic HMC problem with $n + 1$ classes, and a class $A \in \mathcal{S}$ being a subclass of $A, A_1, \ldots, A_n$. Suppose $h_A > h_{A_1}, \ldots, h_{A_n}$, $y_A = 0$, and $y_{A_1}, \ldots, y_{A_n} = 1$. Then, if we use the standard binary cross-entropy loss, we obtain:

$$\mathcal{L} = \mathcal{L}_A + \sum_{i=1}^{n} \mathcal{L}_{A_i}, \qquad \mathcal{L} = -\ln(1 - h_A) - n\ln(h_A), \qquad \frac{\partial \mathcal{L}}{\partial h_A} = \frac{1}{1 - h_A} - \frac{n}{h_A}.$$

Since $y_A = 0$, we would like to get $\frac{\partial \mathcal{L}_A}{\partial h_A} > 0$. However, that is possible only if $h_A > \frac{n}{n+1}$. Let $n = 1$, then we need $h_A > 0.5$, while if $n = 10$, we need $h_A > 10/11 \sim 0.91$. On the contrary, if we use MCLoss, we obtain:

$$\text{MCLoss} = \text{MCLoss}_A + \sum_{i=1}^{n} \text{MCLoss}_{A_i}, \quad \text{MCLoss} = -\ln(1 - h_A) + \sum_{i=1}^{n} \text{MCLoss}_{A_i}, \quad \frac{\partial \text{MCLoss}}{\partial h_A} = \frac{1}{1 - h_A}.$$

Thus, no matter the value of $h_A$, we get $\frac{\partial \text{MCLoss}_A}{\partial h_A} > 0$.

Finally, thanks to both MCM and MCLoss, C-HMCNN($h$) has the ability of delegating the prediction on a class $A$ to one of its subclasses.

**Definition 4.3** (Delegate). Let $\mathcal{S} = \{A_1, \ldots, A_n\}$ be a set of hierarchically structured classes. Let $x \in \mathbb{R}^D$ be a datapoint. Let $h_{A_1}, \ldots, h_{A_n} \in [0, 1]$ be the outputs of a neural network $h$ given the input $x$. Let $\text{MCM}_{A_1}, \ldots, \text{MCM}_{A_n}$ be defined as in Eq. (3). Consider a class $A_i \in \mathcal{S}$ and a class $A_j \in \mathcal{D}_{A_i}$ with $i \neq j$. Then, C-HMCNN($h$) *delegates* the prediction on $A_i$ to $A_j$ for $x$, if $\text{MCM}_{A_i} = h_{A_j}$ and $h_{A_j} > h_{A_i}$.

Consider the basic case in Section 3 and the figures in the last column of Figure 1. Thanks to MCM and MCLoss, C-HMCNN($h$) behaves as expected: it delegates the prediction on $B$ to $A$ for (i) 0% of the points in $A$ when $R_1 \cap R_2 = R_1$ (top figure), (ii) 100% of the points in $A$ when $R_1 \cap R_2 = \emptyset$ (middle figure), and (iii) 85% of the points in $A$ when $R_1$ and $R_2$ are as in the bottom figure.

# 5 Experimental analysis

In this section, we first discuss how to effectively implement C-HMCNN($h$), leveraging GPU architectures. Then, we present the experimental results of C-HMCNN($h$), first considering two synthetic experiments, and then on 20 real-world datasets for which we compare with current state-of-the-art models for HMC problems. Finally, ablation studies highlight the positive impact of both MCM and MCLoss on C-HMCNN($h$)'s performance.[2]

The metric that we use to evaluate models is the area under the average precision and recall curve $AU(\overline{PRC})$. The $AU(\overline{PRC})$ is computed as the area under the average precision recall curve, whose points $(\overline{Prec}, \overline{Rec})$ are computed as:

$$\overline{Prec} = \frac{\sum_{i=1}^{n} \text{TP}_i}{\sum_{i=1}^{n} \text{TP}_i + \sum_{i=1}^{n} \text{FP}_i} \qquad \overline{Rec} = \frac{\sum_{i=1}^{n} \text{TP}_i}{\sum_{i=1}^{n} \text{TP}_i + \sum_{i=1}^{n} \text{FN}_i},$$

where $\text{TP}_i$, $\text{FP}_i$, and $\text{FN}_i$ are the number of true positives, false positives, and false negatives for class $i$, respectively. $AU(\overline{PRC})$ has the advantage of being independent from the threshold used to predict when a datapoint belongs to a particular class (which is often heavily application-dependent) and is the most used in the HMC literature [3, 32, 33].

## 5.1 GPU implementation

For readability, $\text{MCM}_A$ and $\text{MCLoss}_A$ have been defined for a specific class $A$. However, it is possible to compute MCM and MCLoss for all classes in parallel, leveraging GPU architectures.

Let $H$ be an $n \times n$ matrix obtained by stacking $n$ times the $n$ outputs of the bottom module $h$ of C-HMCNN($h$). Let $M$ be an $n \times n$ matrix such that, for $i, j \in \{1, \ldots, n\}$, $M_{ij} = 1$ if $A_j$ is a subclass of $A_i$ ($A_j \in \mathcal{D}_{A_i}$), and $M_{ij} = 0$, otherwise. Then,

$$\text{MCM} = \max(M \odot H, \dim = 1),$$

where $\odot$ represents the Hadamard product, and given an arbitrary $p \times q$ matrix $Q$, $\max(Q, \dim = 1)$ returns a vector of length $p$ whose $i$-th element is equal to $\max(Q_{i1}, \ldots, Q_{iq})$. For MCLoss, we can use the same mask $M$ to modify the standard binary cross-entropy loss (BCELoss) that can be found in any available library (e.g., PyTorch). In detail, let $y$ be the ground-truth vector, $[h_{A_1}, \ldots, h_{A_n}]$ be the output vector of $h$, $\bar{h} = y \odot [h_{A_1}, \ldots, h_{A_n}]$, $\bar{H}$ be the $n \times n$ matrix obtained by stacking $n$ times the vector $\bar{h}$. Then,

$$\text{MCLoss} = \text{BCELoss}(((1-y) \odot \text{MCM}) + (y \odot \max(M \odot \bar{H}, \dim = 1)), y).$$

## 5.2 Synthetic experiment 1

Consider the generalization of the experiment in Section 4 in which we started with $R_1$ outside $R_2$ (as in the second row of Figure 1), and then moved $R_1$ towards the centre of $R_2$ (as in the first row of Figure 1) in 9 uniform steps. The last row of Figure 1 corresponds to the fifth step, i.e., $R_1$ was halfway.

This experiment is meant to show how the performance of C-HMCNN($h$), $f^+$, and $g^+$ as in Section 3 vary depending on the relative positions of $R_1$ and $R_2$. Here, $f$, $g$, and $h$ were implemented and trained as in Section 3. For each step, we run the experiment 10 times,[3] and we plot the mean $AU(\overline{PRC})$ together with the standard deviation for C-HMCNN($h$), $f^+$, and $g^+$ in Figure 2.

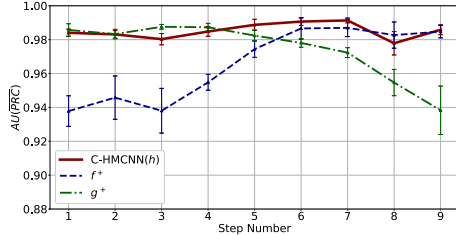

As expected, Figure 2 shows that $f^+$ performed poorly in the first three steps when $R_1 \cap R_2 = \emptyset$, it then started to perform better at step 4 when

Figure 2: Mean $AU(\overline{PRC})$ with standard deviation of C-HMCNN($h$), $f^+$, and $g^+$ for each step.

$R_1 \cap R_2 \notin \{R_1, \emptyset\}$, and it performed well from step 6 when $R_1$ overlaps significantly with $R_2$ (at least 65% of its area). Conversely, $g^+$ performed well on the first five steps, and its performance started decaying from step 6. C-HMCNN($h$) performed well at all steps, as expected, showing robustness with respect to the relative positions of $R_1$ and $R_2$.

## 5.3 Synthetic experiment 2

In order to prove the importance of using MCLoss instead of $\mathcal{L}$, in this experiment we compare two models: (i) our model C-HMCNN($h$), and (ii) $h + \text{MCM}$, i.e., $h$ with MCM built on top and trained with the standard binary cross-entropy loss $\mathcal{L}$. Consider the nine rectangles arranged as in Figure 3 named $R_1, \ldots, R_9$. Assume (i) that we have classes $A_1 \ldots A_9$, (ii) that a datapoint belongs to $A_i$ if it belongs to the $i$-th rectangle, and (iii) that $A_5$ (resp., $A_3$) is an ancestor (resp., descendant) of every class. Thus, all points in $R_3$ belong to all classes, and if a datapoint belongs to a rectangle, then it also belongs to class $A_5$. The datasets consisted of 5000 (50/50 train test split) datapoints sampled from a uniform distribution over $[0, 1]^2$.

Let $h$ be a feedforward neural network with a single hidden layer with 7 neurons. We train both $h + \text{MCM}$ and C-HMCNN($h$) for 20k epochs using Adam optimization with learning rate $10^{-2}$ ($\beta_1 = 0.9, \beta_2 = 0.999$). As expected, the average $AU(\overline{PRC})$ (and standard deviation) over 10 runs for $h + \text{MCM}$ trained with $\mathcal{L}$ is 0.938 (0.038), while $h + \text{MCM}$ trained with MCLoss (C-HMCNN($h$)) is 0.974 (0.007). Notice that not only $h + \text{MCM}$ performs worse, but also, due to the convergence to bad local optima, the

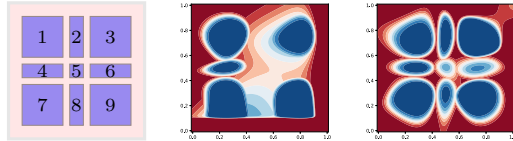

Figure 3: From left to right: (i) rectangles disposition, (ii) decision boundaries for $A_5$ of $h + \text{MCM}$ trained with $\mathcal{L}$, and (iii) decision boundaries for $A_5$ of C-HMCNN($h$).

standard deviation obtained with $h + \text{MCM}$ is 5 times higher than the one of C-HMCNN($h$): the (min, median, max) $AU(\overline{PRC})$ for $h + \text{MCM}$ are $(0.871, 0.945, 0.990)$, while for C-HMCNN($h$) are $(0.964, 0.975, 0.990)$. Since $h + \text{MCM}$ presents a high standard deviation, the figure shows the decision boundaries of the 6th best performing networks for class $A_5$.

Table 1: Summary of the 20 real-world datasets. Number of features ($D$), number of classes ($n$), and number of datapoints for each dataset split.

| TAXONOMY | DATASET | $D$ | $n$ | TRAINING | VALIDATION | TEST |
|---|---|---|---|---|---|---|
| FUNCAT (FUN) | CELLCYCLE | 77 | 499 | 1625 | 848 | 1281 |
| FUNCAT (FUN) | DERISI | 63 | 499 | 1605 | 842 | 1272 |
| FUNCAT (FUN) | EISEN | 79 | 461 | 1055 | 529 | 835 |
| FUNCAT (FUN) | EXPR | 551 | 499 | 1636 | 849 | 1288 |
| FUNCAT (FUN) | GASCH1 | 173 | 499 | 1631 | 846 | 1281 |
| FUNCAT (FUN) | GASCH2 | 52 | 499 | 1636 | 849 | 1288 |
| FUNCAT (FUN) | SEQ | 478 | 499 | 1692 | 876 | 1332 |
| FUNCAT(FUN) | SPO | 80 | 499 | 1597 | 837 | 1263 |
| GENE ONTOLOGY (GO) | CELLCYCLE | 77 | 4122 | 1625 | 848 | 1281 |
| GENE ONTOLOGY (GO) | DERISI | 63 | 4116 | 1605 | 842 | 1272 |
| GENE ONTOLOGY (GO) | EISEN | 79 | 3570 | 1055 | 528 | 835 |
| GENE ONTOLOGY (GO) | EXPR | 551 | 4128 | 1636 | 849 | 1288 |
| GENE ONTOLOGY (GO) | GASCH1 | 173 | 4122 | 1631 | 846 | 1281 |
| GENE ONTOLOGY (GO) | GASCH2 | 52 | 4128 | 1636 | 849 | 1288 |
| GENE ONTOLOGY (GO) | SEQ | 478 | 4130 | 1692 | 876 | 1332 |
| GENE ONTOLOGY (GO) | SPO | 80 | 4166 | 1597 | 837 | 1263 |
| TREE | DIATOMS | 371 | 398 | 1085 | 464 | 1054 |
| TREE | ENRON | 1000 | 56 | 692 | 296 | 660 |
| TREE | IMCLEF07A | 80 | 96 | 7000 | 3000 | 1006 |
| TREE | IMCLEF07D | 80 | 46 | 7000 | 3000 | 1006 |

## 5.4 Comparison with the state of the art

We tested our model on 20 real-world datasets commonly used to compare HMC systems (see, e.g., [3, 23, 32, 33]): 16 are functional genomics datasets [9], 2 contain medical images [13], 1 contains images of microalgae [14], and 1 is a text categorization dataset [17].[4] The characteristics of these datasets are summarized in Table 1. These datasets are particularly challenging, because their number of training samples is rather limited, and they have a large variation, both in the number of features (from 52 to 1000) and in the number of classes (from 56 to 4130). We applied the same preprocessing to all the datasets. All the categorical features were transformed using one-hot encoding. The missing values were replaced by their mean in the case of numeric features and by a vector of all zeros in the case of categorical ones. All the features were standardized.

We built $h$ as a feedforward neural network with two hidden layers and ReLU non-linearity. To prove the robustness of C-HMCNN($h$), we kept all the hyperparameters fixed except the hidden dimension and the learning rate used for each dataset, which are given in Appendix B and were optimized over the validation sets. In all experiments, the loss was minimized using Adam optimizer with weight decay $10^{-5}$, and patience 20 ($\beta_1 = 0.9$, $\beta_2 = 0.999$). The dropout rate was set to 70% and the batch size to 4. As in [33], we retrained C-HMCNN($h$) on both training and validation data for the same number of epochs, as the early stopping procedure determined was optimal in the first pass.

For each dataset, we run C-HMCNN($h$), Clus-Ens [28], and HMC-LMLP [7] 10 times, and the average $AU(\overline{PRC})$ is reported in Table 2. For simplicity, we omit the standard deviations, which for C-HMCNN($h$) are in the range $[0.5 \times 10^{-3}, 2.6 \times 10^{-3}]$, proving that it is a very stable model. As reported in [23], Clus-Ens and HMC-LMLP are the current state-of-the-art models with publicly available code. These models were run with the suggested configuration settings on each dataset.[5] The results are shown in Table 2, left side. On the right side, we show the results of HMCN-R and HMCN-F directly taken from [33], since the code is not publicly available. We report the results of both systems, because, while HMCN-R has worse results than HMCN-F, the amount of parameters of the latter grows with the number of hierarchical levels. As a consequence, HMCN-R is much lighter in terms of total amount of parameters, and the authors advise that for very large hierarchies, HMCN-R is probably a better choice than HMCN-F considering the trade-off performance vs. computational

Table 2: Comparison of C-HMCNN($h$) with the other state-of-the-art models. The performance of each system is measured as the $AU(\overline{PRC})$ obtained on the test set. The best results are in bold.

| Dataset | C-HMCNN($h$) | HMC-LMLP | Clus-Ens | HMCN-R | HMCN-R |
|---|---|---|---|---|---|
| Cellcycle FUN | **0.255** | 0.207 | 0.227 | 0.247 | 0.252 |
| Derisi FUN | **0.195** | 0.182 | 0.187 | 0.189 | 0.193 |
| Eisen FUN | **0.306** | 0.245 | 0.286 | 0.298 | 0.298 |
| Expr FUN | **0.302** | 0.242 | 0.271 | 0.300 | 0.301 |
| Gasch1 FUN | **0.286** | 0.235 | 0.267 | 0.283 | 0.284 |
| Gasch2 FUN | **0.258** | 0.211 | 0.231 | 0.249 | 0.254 |
| Seq FUN | **0.292** | 0.236 | 0.284 | 0.290 | 0.291 |
| Spo FUN | **0.215** | 0.186 | 0.211 | 0.210 | 0.211 |
| Cellcycle GO | **0.413** | 0.361 | 0.387 | 0.395 | 0.400 |
| Derisi GO | **0.370** | 0.343 | 0.361 | 0.368 | 0.369 |
| Eisen GO | **0.455** | 0.406 | 0.433 | 0.435 | 0.440 |
| Expr GO | 0.447 | 0.373 | 0.422 | 0.450 | **0.452** |
| Gasch1 GO | **0.436** | 0.380 | 0.415 | 0.416 | 0.428 |
| Gasch2 GO | 0.414 | 0.371 | 0.395 | 0.463 | **0.465** |
| Seq GO | 0.446 | 0.370 | 0.438 | 0.443 | **0.447** |
| Spo GO | **0.382** | 0.342 | 0.371 | 0.375 | 0.376 |
| Diatoms | **0.758** | - | 0.501 | 0.514 | 0.530 |
| Enron | **0.756** | - | 0.696 | 0.710 | 0.724 |
| Imclef07a | **0.956** | - | 0.803 | 0.904 | 0.950 |
| Imclef07d | **0.927** | - | 0.881 | 0.897 | 0.920 |
| Average Ranking | 1.25 | 5.00 | 3.93 | 2.93 | 1.90 |

cost [33]. Note that the number of parameters of C-HMCNN($h$) is independent from the number of hierarchical levels.

As reported in Table 2, C-HMCNN($h$) has the greatest number of wins (it has the best performance on all datasets but 3) and best average ranking (1.25). We also verified the statistical significance of the results following [11]. We first executed the Friedman test, obtaining p-value $4.26 \times 10^{-15}$. We then performed the post-hoc Nemenyi test, and the resulting critical diagram is shown in Figure 4, where the group of methods that do not differ significantly (significance level 0.05) are connected through a horizontal line. The Nemenyi test is powerful enough to conclude that there is a statistical significant difference between the performance of C-HMCNN($h$) and all the other models but HMCN-F. Hence, following [11, 2], we compared C-HMCNN($h$)

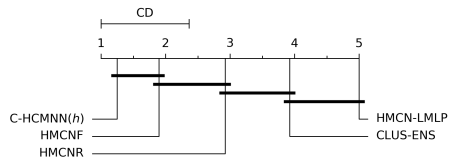

Figure 4: Critical diagram for the Nemenyi's statistical test.

and HMCN-F using the Wilcoxon test. This test, contrarily to the Friedman test and the Nemenyi test, takes into account not only the ranking, but also the differences in performance of the two algorithms. The Wilcoxon test allows us to conclude that there is a statistical significant difference between the performance of C-HMCNN($h$) and HMCN-F with p-value of $6.01 \times 10^{-3}$.

## 5.5 Ablation studies

To analyze the impact of both MCM and MCLoss, we compared the performance of C-HMCNN($h$) on the validation set of the FunCat datasets against the performance of $h^+$, i.e., $h$ with the post-processing as in [7] and [15] and $h$+MCM, i.e., $h$ with MCM built on top. Both these models were trained using the standard binary cross-entropy loss. As it can be seen in Table 3, MCM by itself already helps to improve the performances on all datasets but Derisi, where $h^+$ and $h$+MCM have the same performance. However, C-HMCNN($h$), by exploiting both MCM and MCLoss, always outperforms $h^+$ and $h$+MCM. In Table 3, we also report after how many epochs the algorithm stopped training in average. As it can be seen, even though C-HMCNN($h$) and $h$+MCM need more epochs than $h^+$, the numbers are still comparable.

Table 3: Impact of MCM and MCM+MCLoss on the performance measured as $AU(\overline{PRC})$ and on the total number of epochs for the validation set of the Funcat datasets.

| Dataset | $h^+$ | | $h$+MCM | | C-HMCNN($h$) | |
|---|---|---|---|---|---|---|
| | $AU(\overline{PRC})$ | Epochs | $AU(\overline{PRC})$ | Epochs | $AU(\overline{PRC})$ | Epochs |
| CELLCYCLE | 0.220 | 74 | 0.229 | 108 | **0.232** | 106 |
| DERISI | 0.179 | 58 | 0.179 | 66 | **0.182** | 67 |
| EISEN | 0.262 | 76 | 0.271 | 107 | **0.285** | 110 |
| EXPR | 0.246 | 14 | 0.265 | 19 | **0.270** | 20 |
| GASCH1 | 0.239 | 28 | 0.258 | 42 | **0.261** | 38 |
| GASCH2 | 0.221 | 103 | 0.234 | 132 | **0.235** | 131 |
| SEQ | 0.245 | 8 | 0.269 | 13 | **0.274** | 13 |
| SPO | 0.186 | 103 | 0.189 | 117 | **0.190** | 115 |
| AVERAGE RANKING | 2.94 | | 2.06 | | 1.00 | |

# 6 Related work

HMC problems are a generalization of hierarchical classification problems, where the labels are hierarchically organized, and each datapoint can be assigned to one path in the hierarchy (e.g., [10, 26, 30]). Indeed, in HMC problems, each datapoint can be assigned multiple paths in the hierarchy.

In the literature, HMC methods are traditionally divided into local and global approaches [29]. Local approaches decompose the problem into smaller classification ones, and then the solutions are combined to solve the main task. Local approaches can be further divided based on the strategy that they deploy to decompose the main task. If a method trains a different classifier for each level of the hierarchy, then we have a *local classifier per level* as in [5-7, 21, 35]. The works [5-7] are extended by [33], where HMCN-R and HMCN-F are presented. Since HMCN-R and HMCN-F are trained with both a local loss and a global loss, they are considered hybrid local-global approaches. If a method trains a classifier for each node of the hierarchy, then we have a *local classifier per node*. In [8], a linear classifier is trained for each node with a loss function that captures the hierarchy structure. On the other hand, in [15], one multi-layer perceptron for each node is deployed. A different approach is proposed in [3], where kernel dependency estimation is employed to project each label to a low-dimensional vector. To preserve the hierarchy structure, a generalized condensing sort and select algorithm is developed, and each vector is then learned singularly using ridge regression. Finally, if a method trains a different classifier per parent node in the hierarchy, then we have a *local classifier per parent node*. For example, [18] proposes to train a model for each sub-ontology of the Gene Ontology, combining features automatically learned from the sequences and features based on protein interactions. In [34], instead, the authors try to solve the overfitting problem typical of local models by representing the correlation among the labels by the label distribution, and then training each local model to map datapoints to label distributions. Global methods consist of single models able to map objects with their corresponding classes in the hierarchy as a whole. A well-known global method is CLUS-HMC [32], consisting of a single predictive clustering tree for the entire hierarchy. This work is extended in [28], where Clus-Ens, an ensemble of CLUS-HMC, is proposed. In [22], a neural network incorporating the structure of the hierarchy in its architecture is proposed. While this network makes predictions that are coherent with the hierarchy, it also makes the assumption that each parent class is the union of the children. In [4], the authors propose a "competitive neural network", whose architecture replicates the hierarchy.

# 7 Summary and outlook

In this paper, we proposed a new model for HMC problems, called C-HMCNN($h$), which is able to (i) leverage the hierarchical information to learn when to delegate the prediction on a superclass to one of its subclasses, (ii) produce predictions coherent by construction, and (iii) outperfom current state-of-the-art models on 20 commonly used real-world HMC benchmarks. Further, its number of parameters does not depend on the number of hierarchical levels, and it can be easily implemented on GPUs using standard libraries. In the future, we will use as $h$ an interpretable model (see, e.g., [19]), and study how MCM and MCLoss can be modified to improve the interpretability of C-HMCNN($h$).

## Broader Impact

In this paper, we proposed a novel model that is shown to outperform the current state-of-the-art models on commonly used HMC benchmarks. We expect our approach to have a large impact on the research community not only because of its positive results but also because it is relatively easy to implement and test using standard libraries, and the code os publicly available. From the application perspective, given the generality of the approach, it is impossible to foresee all the possible impacts in all the different application domains where HMC problems arise. We thus focus on functional genomics, which is the application domain most benchmarks come from.

The goal in functional genomics is to describe the functions and interactions of genes and their products, RNA and proteins. As stated in [23, 25], in recent years, the generation of proteomic data has increased substantially, and annotating all sequences is costly and time-consuming, making it often unfeasible. It is thus necessary to develop methods (like ours) that are able to automatize this process. Having better models for such a task may unlock many possibilities. Indeed, it may (i) allow to better understand the role of proteins in disease pathobiology, (ii) help determine the function of metagenomes offering possibilities to discover novel genes and novel biomolecules, and (iii) facilitate finding drug targets, which is crucial to the success of mechanism-based drug discovery.

## Acknowledgments and Disclosure of Funding

We would like to thank Francesco Giannini and Marco Gori for useful discussions, and Maria Kiourlappou for her feedback on the broader impact section. Eleonora Giunchiglia is supported by the EPSRC under the grant EP/N509711/1 and by an Oxford-DeepMind Graduate Scholarship. This work was also supported by the Alan Turing Institute under the EPSRC grant EP/N510129/1 and by the AXA Research Fund. We also acknowledge the use of the EPSRC-funded Tier 2 facility JADE (EP/P020275/1) and GPU computing support by Scan Computers International Ltd.

## Footnotes

[1]By definition, $A \in \mathcal{D}_A$.

[2]Link: https://github.com/EGiunchiglia/C-HMCNN/

[3]All subfigures in Figure 1 correspond to the decision boundaries of $f$, $g$, and $h$ in the first of the 10 runs.

[4]Links: https://dtai.cs.kuleuven.be/clus/hmcdatasets and http://kt.ijs.si/DragiKocev/PhD/resources

[5]We also ran the code from [22]. However, we obtained very different results from the ones reported in the paper. Similar negative results are also reported in [23].

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
