[Supplementary Material]

# A  Decision Boundaries

Figure 5: Figure 5a: Decision boundaries for class $B$ of $f^+$ when $R_1 \cap R_2 = \emptyset$. Figure 5b: Decision boundaries for class $B$ of $f^+$ when $R_1 \cap R_2 = R_1$. Figure 5c: Decision boundaries for class $B$ of $f^+$ when $R_1 \cap R_2 \notin \{\emptyset, R_1\}$.

Figure 6: Figure 6a: Decision boundaries for class $B$ of $g^+$ when $R_1 \cap R_2 = \emptyset$. Figure 6b: Decision boundaries for class $B$ of $g^+$ when $R_1 \cap R_2 = R_1$. Figure 6c: Decision boundaries for class $B$ of $g^+$ when $R_1 \cap R_2 \notin \{\emptyset, R_1\}$.

Figure 7: Figure 7a: Decision boundaries for class $B$ of C-HMCNN($h$) when $R_1 \cap R_2 = \emptyset$. Figure 7b: Decision boundaries for class $B$ of C-HMCNN($h$) when $R_1 \cap R_2 = R_1$. Figure 7c: Decision boundaries for class $B$ of C-HMCNN($h$) when $R_1 \cap R_2 \notin \{\emptyset, R_1\}$.

# B  Experimental Analysis Details

In this section, we provide more details about the conducted experimental analysis. As stated in the paper, across the different experiments, we kept all hyperparameters fixed with the exception of the hidden dimension and the learning rate, which are reported in the first two columns of Table 4. The other hyperparameters were determined by searching the best hyperparameters configuration on the Funcat datasets; we then took the configuration that led to the best results on the highest

Table 4: Hidden dimension used for each dataset, learning rate used for each dataset, and average inference time per batch in milliseconds (ms). Average computed over 500 batches for each dataset.

| DATASET | Hidden Dimension | Learning Rate | Time per batch (ms) |
|---|---|---|---|
| CELLCYCLE FUN | 500 | $10^{-4}$ | 2.0 |
| DERISI FUN | 500 | $10^{-4}$ | 2.0 |
| EISEN FUN | 500 | $10^{-4}$ | 1.7 |
| EXPR FUN | 1000 | $10^{-4}$ | 1.9 |
| GASCH1 FUN | 1000 | $10^{-4}$ | 2.0 |
| GASCH2 FUN | 500 | $10^{-4}$ | 2.8 |
| SEQ FUN | 2000 | $10^{-4}$ | 2.0 |
| SPO FUN | 250 | $10^{-4}$ | 1.6 |
| CELLCYCLE GO | 1000 | $10^{-4}$ | 2.4 |
| DERISI GO | 500 | $10^{-4}$ | 2.5 |
| EISEN GO | 500 | $10^{-4}$ | 3.4 |
| EXPR GO | 4000 | $10^{-5}$ | 3.9 |
| GASCH1 GO | 500 | $10^{-4}$ | 2.5 |
| GASCH2 GO | 500 | $10^{-4}$ | 2.8 |
| SEQ GO | 9000 | $10^{-5}$ | 2.6 |
| SPO GO | 500 | $10^{-4}$ | 3.3 |
| DIATOMS | 2000 | $10^{-5}$ | 2.0 |
| ENRON | 1000 | $10^{-5}$ | 3.6 |
| IMCLEF07A | 1000 | $10^{-5}$ | 3.4 |
| IMCLEF07D | 1000 | $10^{-5}$ | 2.9 |

number of datasets. The hyperparameter values taken in consideration were: (i) learning rate: $[10^{-3}, 10^{-4}, 10^{-5}]$, (ii) batch size: $[4, 64, 256]$, (iii) dropout: $[0.6, 0.7]$, and (iv) weight decay: $[10^{-3}, 10^{-5}]$. Concerning the hidden dimension, we took into account all possible dimensions from 250 to 2000 with step equal to 250, and from 2000 to 10000 with step 1000. The last column of Table 4 shows the average inference time per batch in milliseconds. The average is computed over 500 batches for each dataset. All experiments were run on an Nvidia Titan Xp with 12 GB memory.