[Reviews · NeurIPS 2020]

Review 1

Summary and Contributions: The paper introduces a new approach for Hierarchical multi-label classification problems. In such problems, in addition to the challenges faced by multi-label classification problems, there is an additional constraint that requires predictions to be consistent with the hierarchy. Their approach leverages the hierarchy information to make predictions that are consistent by construction by learning to delegate the prediction on a superclass to one of the subclasses. They introduce a novel loss function for the same. In the experimental results they show that they outperform state of the art models on commonly used real world benchmarks.

Strengths: The paper deals with a problem with a lot of significance and relevance to real world applications. The experimental results are quite consistent with their claims. They have provided information about the experiments in good detail. There are also novel aspects in the paper with a seemingly broader impact.

Weaknesses: The motivation for the choice of loss function is explained through what appears to be a specific example - perhaps some more motivation could be given about how such an example could be a common occurrence.

Correctness: To the best of my knowledge their claims and methodology appear to be correct.

Clarity: The paper is well written and the explanations starting from the basic case and moving to the general case makes the paper more understandable.

Relation to Prior Work: They have discussed a number of prior works and how their approach differs, but I lack the subject knowledge in this particular problem to know if they have addressed all the previous contributions.

Reproducibility: Yes

Additional Feedback:


Review 2

Summary and Contributions: This paper discusses the problem setting of hierarchical multi-label classification where predictions have to be consistent with the hierarchy. As central idea, the authors present a neural network architecture which guarantees that child-labels are only predicted positive when their parent label is predicted positive. The architecture is in essence a binary relevance decomposition of the problem, but the loss function is adjusted so that hierarchical constraints are obeyed. The method is evaluated on classical benchmarks, and compared to other neural network implementations.

Strengths: - the presented method is novel - the presented method is well described and intuitively visualized

Weaknesses: - hierarchical multi-label classification is a heavily investigated setting in machine learning, so it is hard to introduce a method that outperforms the state-of-the-art - the experimental results are questionable - the need for this method is not fully demonstrated

Correctness: The presented method is relatively simple, but appealing. The authors do a good job in explaining the main characteristics of their method, and the situations where it might lead to a good performance. The experiments on the synthetic datasets are nice as an illustration. The experiments on the benchmarks are not very convincing. Those benchmarks circulate already for a long time in the ML community. Before the deep learning era, they have been extensively used to benchmark newly-developed kernel and tree-based methods. It is hard to believe that neural networks can outperform more classical methods on such datasets, because the samples size are typically small. It is commonly known that the power of neural networks is the ability of feature learning, but this is not at all exploited in this paper. The authors use traditional fully-connected networks, in combination with the same hand-crafted features as in older papers. What is the added value of using a neural network here? For datasets with a sufficiently large sample size, it would be useful to experiment with 2D convolutions (image datasets) and 1D convolutions (genomics) datasets.

Clarity: Yes, the paper is very well written.

Relation to Prior Work: To my opinion, the related work section remains quite vague concerning the exact problem setting tackled in this paper. Perhaps start this section with a paragraph that discusses methods for exactly the same problem setting, including older methods, such as kernel-based or tree-based methods. Then, in a new paragraph, discuss papers that analyze slightly different problem settings. For example, there is a lot of work in which specific loss functions for hierarchical multi-label classification is discussed. Several loss functions, such a the tree distance loss, try to penalize predictions that are inconsistent with the hierarchy. These papers handle a slightly different problem setting because consistency is not enforced but only favored in those case. However, such papers are in fact ignored in the related work section.

Reproducibility: Yes

Additional Feedback: I liked the toy examples, which illustrate in which situations the algorithm might work well. I would argue that the setting where a child label corresponds to a region of the space that is fully included by the bigger region of its parent label is most logical. For hierarchical multi-label classification problems, child labels can often semantically be seen as specializations of their parent labels. This can indeed be translated to the feature space, but I would argue that in practice this happens in a slightly different way. Child labels often have the same features active as their parents, e.g. certain shape features in image datasets or sequence motifs in genomics datasets. In addition, they have additional features active that their parents don't have. The presented toy problems mimic this to a certain extent, but one could develop slightly different toy problems that analyze this more correctly. It would be interesting to see whether the presented architecture can cope well with such situations. Another concern I have is related to the proposed loss. Due to the max operator, this loss becomes a complex function to optimize. Is there a danger to get stuck in bad local optima? --------------------------------- Review update after discussion phase: ---------------------------------- The notion "consistency" is not so well chosen by the authors. What they mean with consistency is that the predictions need to obey the following rule: a child node can only be positive when its ancestors are positive. So, this has nothing to do with Fisher consistency. However, I don't see this as a reason for rejecting the paper, because the authors could simply replace the word consistency by something else in their camera-ready version. I forgot to mention this in my review. Therefore, we state more clearly that we expect the word consistency to be replaced if the paper gets accepted. I enjoyed reading this paper. It deals with a specific but interesting problem setting, and the provided solution makes sense. The authors don't have theoretical results in the sense of Theorems, but they do provide a discussion of the scenarios where their method might be useful. I don't think that a good Neurips paper needs to have theorems, and many good papers without theorems got accepted in previous years. My concerns with this paper were mostly related to the experiments, but in that regard the rebuttal is somewhat convincing and changing my opinion on the positive side. A question that still remains unanswered to me is why neural networks outperform tree-based methods on these "quite simple" datasets. I still don't see what the advantage of neural networks would be here.


Review 3

Summary and Contributions: The authors propose a novel approach, C-HMCNN(h), for Hierarchical multi-label classification problem, which is a sub-network applied to multi-label classification network. And for this, max constraint loss (MCLoss) is proposed to optimize and exploit the hierarchy constraint during training. The experiments were conducted to show the superior performance of C-HMCNN(h) with other state-of-the-art models.

Strengths: In the process of model construction, full experiments are conducted from a special cases (only two classes A and B). A possible effective method is obtained by analyzing the results. Then it is generalized to the general situation. An example calculation proves the effectiveness of MCLoss. The whole model has a certain interpretability and rationality. The writing is fluent, the thinking is clear, the symbol description and method are detailed and easy to understand.

Weaknesses: 1. The novelty is limited. Using the max(h_A, h_B) instead of h_B during training has been proposed by other methods such as DeepGO. The further improvement of C-HMCNN(h) is using a slightly different loss function MCMLoss. But the performance of MCMLoss + MCM is only a little higher than MCM. And the performance of C-HMCNN(h) is slightly higher than SOTA method HMCN-F (sometimes even lower). 2. The authors should give more insights why MCLoss is better than standard binary cross-entropy (L). One example may not be enough to explain the problem. 3. In the experiments, it would be better to give the detail of the hierarchical DAG (the number of edges in the DAG). The authors stated that "its number of parameters does not depend on the number of hierarchical levels". In this problem, the number of hierarchical levels is not the main problem, but the number of edges in the DAG is.

Correctness: Sounds correct

Clarity: Yes

Relation to Prior Work: Yes

Reproducibility: Yes

Additional Feedback:


Review 4

Summary and Contributions: This paper addresses the problem of hierarchical multi-label classification, which is known to be the most challenging type of classification problem. Basically, the authors propose two main novelties for targeting the problem. First, the authors propose a simple yet effective way of controlling the hierarchical constraint (which states that all superclasses of a given class must be predicted if a given class is predicted), which is a simple module they call max constraint module (MCM). The MCM defines the prediction of a given superclass as the largest value between the superclass prediction and all its subclasses predictions. Therefore, it is now impossible to violate a hierarchical constraint. The second contribution is a simple modification over the binary cross-entropy loss, by simply incorporating the MCM predictions for each positive class (and its complement when that class is not supposed to be predicted).

Strengths: What I really like about this paper is its sheer simplicity. The ideas are dead simple, yet very effective! The authors manage to achieve state-of-the-art results (for virtually every major benchmark dataset used in the literature of HMC) with a simple fully-connected network (single hidden layer). I think we should appreciate the fact that those simple ideas work so well in this area. Most papers on HMC propose very complex strategies for addressing the problem, and results are usually not better than those of predictive clustering trees, from Vens et al. Only by 2018 that the area seems to have advanced with the paper of Wehrmann and Barros, on ICML 2018, which introduced some non trivial architectures that made the networks dependent on the number of hierarchical levels of the class hierarchy. Now, we can see that a somewhat much simple approach is even more effective.

Weaknesses: Perhaps the weakness of this paper is not to present the hyperparameter optimization process in detail. I know the authors have left some details to the supplementary material, but I do miss some details on how those hidden neurons and learning rates were defined. Did you optimize them over the validation sets? We've got to be careful with those datasets, since they are overly used in the literature. We should not perform any kind of hyperparameter optimization over the test sets.

Correctness: Yes, the methodology seems to be sound.

Clarity: Yes, the paper is very well written and easy to follow.

Relation to Prior Work: Yes, the main related papers are properly cited, and there is a clear discussion on the differences of this novel approach to prior work.

Reproducibility: Yes

Additional Feedback: I do like this paper very much. Very simple and effective ideas. The area of HMC seems to be once again gaining some interest after a long period of stagnation. My comments to the authors are mainly from an interested reader of the HMC literature. Have you tried to see what would happen if you modify the approach proposed by Wehrmann and Barros (2018), by plugging your novel loss function? Perhaps we would find out it would overfit due to the more complex architectures proposed in that paper, but I'm not very sure on that. I am very curious on whether your loss function could boost the networks proposed in that work, and also other optimization-based approaches.


Review 5

Summary and Contributions: Authors propose a new algorithm for the problem of Hierarchical Multi-label Classification (HMC) characterized by using: - constraint layer, on top of neural network for the underlying multi-label classification problem (with output for each existing class), that ensures the prediction is consistent with hierarchy constraint (prediction for parent class must be higher or equal to the prediction of its child class). - loss function called max constraint loss (MCLoss) extending BCELoss and allowing for delegating prediction on a class to its subclasses. Authors present how to efficiently implement MCLoss on GPU and conduct the experimental study, where they demonstrate the superiority of their method over SOTA.

Strengths: The proposed is clearly explained, and the motivation behind it is nicely presented. Empirical evaluation on synthetic data, comparison to SOTA, and ablation study are present. There is not much more in the paper. However, the proposed method is simple and seems to be effective. It is also general and can be used with many types of underling multi-label classifier, so I believe it is a relevant contribution.

Weaknesses: I do not see the evident weaknesses in this work.

Correctness: Yes, I found no mistakes. My only concern is with empirical results in Tables 2 and 3 for C-HMCNN (proposed method). The results are different between tables, I would expect them to be the same.

Clarity: The first part of the paper is easy to read, the proposed method is nicely explained. However, the second part focusing on experimental analysis is a bit confusing. It lacks an explanation of compared methods that are commented in the Related Works section at the end of the paper. Still, some of the methods are not clearly addressed and described. Authors should use names of the methods in the text more instead of just citation numbers.

Relation to Prior Work: Discussion about previous works is a bit scattered over the paper and it does not place the proposed algorithm in the landscape of related works. The differences and similarities between the authors' work and SOTA are not clearly stated.

Reproducibility: Yes

Additional Feedback: - Why the results of the ablation study presented in Table 3 for C-HMCNN are different than in Table 2? - I suggest moving the Related Work section at the beginning of the paper and rewrite it to address relevant methods more clearly. - In the General Case section, I would add a small reminder that in introduced notation class is its own subclass, and D is a set of all subclasses (on all levels). I believe it will help some readers. - The equation defining evaluation measures AU(PRC) would be a nice addition. - In Table 2, using bold to highlight the best results among only the first three methods (authors', and two quite old methods) looks like a small manipulation that may be disliked by some. It is also a bit confusing, taking into account the average ranking given in the last row of the table. I suggest removing underline and just bold the best value in each row. - In Table 3, there is AUC(PR), should it be AU(PRC)?. - Some typos I've noticed: - classifer -> classifier - condesing -> condensing Review update after discussion phase: - I agree with other reviewers that the use of the term "consistency" may be a bit confusing for some readers. - I agree that the paper lacks theoretical results, but I do not think theoretical results are necessary, many papers without them got accepted at NeurIPS. - I still think some of the sections need improvments and clarifications.

[Author Response · NeurIPS 2020]

Many thanks to all reviewers for their useful comments, which will improve the final version of the paper. We are very happy that the novelty, simplicity, effectiveness, significance, and generality of our method has been appreciated. We will address all the comments about the related work, the tables, and the details of the datasets in the final paper.

**@R1: Motivation of loss function? @R2: Slightly different toy problems? @R3: Using max has already been proposed by other methods (e.g., DeepGO). @R3: Why MCLoss is better than BCELoss ($\mathcal{L}$)?** While the max function has already been used, nobody so far has shown how to deploy it *effectively*, which is what the combination of MCM and MCLoss does. Experimentally, the ablation studies in Table 4 show that MCLoss allows for *consistent* higher performance than $\mathcal{L}$. In general, consider a class $A$ with ancestors $A_1 \ldots A_n$ in the hierarchy. The higher $n$ the more likely it is that a neural network (NN) with MCM trained with $\mathcal{L}$ will remain stuck in bad local optima. Indeed, consider a datapoint such that $h_A > h_{A_1} \ldots h_{A_n}$, $y_A = 0$, $y_{A_1} \ldots y_{A_n} = 1$: then $\mathcal{L} = -\ln(1 - h_A) - \sum_{i=1}^{n} \ln(h_A) = -\ln(1 - h_A) - n\ln(h_A)$, while MCLoss $= -\ln(1 - h_A) + \sum_{i=1}^{n} \text{MCLoss}_{A_i}$. Notice that, in the GO datasets, it is common to have $n > 10$, given that the hierarchies have 13 levels, and each class can have more than one parent. To visualize the negative impact of using $\mathcal{L}$ instead of MCLoss, consider the leftmost figure with 9 rectangles named $1 \ldots 9$. Further, assume (i) we have classes $A_1 \ldots A_9$, (ii) that a datapoint belongs to $A_i$ if it belongs to the $i$th rectangle, and (iii) that $A_5$ (resp., $A_3$) is an ancestor (resp., descendant) of every class. Thus, all points in rectangle 3 belong to all classes, and if a datapoint belongs to a rectangle, then it also belongs to class $A_5$.

Figure 1: From left to right: (i) rectangles disposition, (ii) decision boundaries for $A_5$ of $h$ + MCM trained with $\mathcal{L}$, and (iii) decision boundaries for $A_5$ of C-HMCNN($h$).

Let $h$ be a NN with a single hidden layer with 7 neurons. Then, the average $AU(\overline{PRC})$ (and std) over 10 runs for $h$ + MCM trained with $\mathcal{L}$ is 0.938 (0.038), while $h$ + MCM trained with MCLoss (C-HMCNN($h$)) is 0.974 (0.007). Notice that not only $h$ + MCM performs worse, but also, due to the convergence to bad local optima, the std obtained with $h$ + MCM is 5 times higher than the one of C-HMCNN($h$): the (min, median, max) $AU(\overline{PRC})$ for $h$ + MCM are $(0.871, 0.945, 0.990)$ while for C-HMCNN($h$) are $(0.964, 0.975, 0.990)$. The figure shows the decision boundaries of the 6th best perfoming networks for class $A_5$ (given the even number of runs, we could not take the exact median). We will include this example and a general discussion in the paper. Thanks also to R2 for the suggestion on how to create alternative synthetic examples: we will explore such direction in the future.

**@R2: The need for this method is not fully demonstrated. @R2 It is hard to believe that NNs can outperform more classical methods on such datasets.** As pointed out by R1, the paper deals with a problem with a lot of significance and relevance to real-world applications, while we achieved SOTA results on most benchmark datasets used in the literature of HMC. As shown by [2], who established the current SOTA on these datasets, NNs can beat more classical methods. In the paper, we compared ourselves with the current SOTA models (one is Clus-Ens: an ensemble of predictive clustering trees), which have already proved to beat famous tree-based and kernel-based methods.

**@R2: What is the added value of using a NN here?** NNs have shown the ability of performing very well in different scenarios. We wanted to create a model that was broadly applicable and easy to use, and hence NNs were our first choice. Notice though that $h$ can be any model that outputs a probability for each class and can be trained with backpropagation.

**@R2: Is there a danger to get stuck in bad local optima?** Our results show our model is very stable. As stated at line 205 in the paper, the std over 10 runs is very small (in the range $[0.5 \times 10^{-3}, 2.6 \times 10^{-3}]$). This is not surprising: while, in theory, gradient descent can be performed only on differentiable functions, in practice, it performs well also on non-differentiable functions (e.g., on ReLU() and max()) leading to very good performances [1]. Considering the max() function, it is differentiable almost everywhere, and software implementations of NN training (e.g., Adam in Pytorch) usually return one of the one-sided derivatives. This may be heuristically justified by observing that gradient-based optimization on a digital computer is subject to numerical error anyway [1]. Hence, in practice, the non-differentiability of a small set of points does not affect the learning algorithm.

**@R3: Comparative analysis of computation time?** In addition to the inference time per batch in Table 4 (appendix), we will include a table with the training times of each model. Due to lack of space, here, we report the ones for CELLCYCLE FUN (in minutes, averaged over 10 runs): (i) C-HMCNN($h$): $\sim$6m, (ii) Clus-Ens: $\sim$20m, and (iii) HMC-LMLP:$\sim$51m. Since we do not have the code, we could not measure the times for HMCN-R and HMCN-F.

**@R4, R2 Plug in our method on top of [2] and/or CNNs?** We still have to try this; we will do so in future work.

**@R4: Did you optimize them over the validation sets?** Yes, we optimized the hyperparameters on the validation sets. We used the test sets only to report the results. We will make it clearer in the appendix.

**@R5: The results are different between tables, I would expect them to be the same.** We conducted the ablation studies on the validation set (see caption of Table 3), while we report the results (after re-training on training+validation set) on the test set. We will specify this in the caption of Table 2.

[1] I. Goodfellow, Y. Bengio, and A. Courville. *Deep Learning*, pages 188–189. MIT Press, 2016. http://www.deeplearningbook.org.
[2] J. Wehrmann, R. Cerri, and R. C. Barros. Hierarchical multi-label classification networks. In *Proc. of ICML*, 2018.


[Meta-Review · NeurIPS 2020]

The submission introduces a new loss function for hierarchical multi-label classification. The justification of the loss function is purely empirical given in a form of results obtained on an illustrative synthetic example. The learning under this loss can be efficiently performed using GPUs. The introduced algorithm obtains the state-of-the-art results. The reviewers agreed that the paper is clearly written, the loss function well-motivated and interesting, and the results worth publishing. Nevertheless, the paper would certainly gain by extending it by theoretical analysis of the loss function. It would be interesting to learn, for example, what is the Bayes optimal decision for this loss function and see an analysis on how well it behaves in optimization (convexity, smoothness). From this point of view the paper is unfortunately very shallow, but the general idea thought-provoking.